# "It's Not a One-Time Conversation": Australian Parental Views on Supporting Young People in Relation to Pornography Exposure

Sally Burke [ID], Mayumi Purvis, Carol Sandiford [ID] and Bianca Klettke *

School of Psychology, Deakin University, 1 Gheringap Street, Geelong, VIC 3220, Australia;
sallymburke7@gmail.com (S.B.); m.purvis@deakin.edu.au (M.P.); c.sandiford@deakin.edu.au (C.S.)
* Correspondence: bianca.klettke@deakin.edu.au; Tel.: +61-3-9244-6207

**Abstract:** While pornography provides opportunities for sexual exploration for young people, early and easy access also has possible negative implications for young people's behavioural and sexual development. Parental responsibilities concerning their children's consumption of pornography are largely misunderstood. This study explored parental experiences and beliefs about pornography education for young people using a qualitative study (*n* = 8, 6 females, 2 males). Interview data were analysed using a reflexive thematic approach. Results indicated that parents have concerns about the ease of access to pornography and the unmediated ideas it presents. Additionally, parents believe they have a responsibility to educate young people about pornography through having open and honest conversations and providing supervision. Further, parents believe that schools should be doing more to educate young people about pornography. This study extends upon current literature by suggesting that although parents feel well-equipped to communicate with and educate young people about pornography consumption, they lack confidence in their capacities to do this.

**Keywords:** pornography; young people; adolescence; parental concern; sexual education; Australia

## 1. Introduction

As modern life advances, children grow up with access to multiple screens and a web of online content at their constant disposal [1]. Specifically, eighty-seven percent of Australians over the age of 15 are internet users, with 15–17-year-olds being the highest internet users at 98% [2]. While this access to screens and the internet provides countless educational opportunities for young people, it also opens them up to potentially risky online experiences, including easy and covert access to internet pornography [3–7].

The prevalence of both intentional and unintentional pornography exposure varies greatly across the literature [8]. Young people's intentional pornography use ranges from 7% in 10–17-year-olds in a United States study [9] to 59% in 10–12th-grade students in a Taiwanese study [10]. An Australian study found that accidental exposure had occurred in 40% of the children included in the study, as reported by parents [11]. This potential for intentional and accidental exposure heightens parental concerns about the appropriate age for initiating conversations.

### 1.1. Age of First Exposure

The age of first exposure to pornography varies greatly in the literature. A recent New Zealand study found the age of the first encounter at 11.7 years [12], while a study in Spain found the age to be 8 years [13]. In Australia, the age has been found to range from age 11 [14] to 13 years [14,15].

This variation in the age of first exposure could contribute to parental decisions about when to initiate conversations regarding pornography. This can result in parents waiting for issues to manifest after young people have viewed pornography [11]. This is

problematic, as a parental discussion prior to the first encounter would be beneficial to young people's development [16] and can help to shape young peoples' attitudes towards pornography [12,17], which we discuss in a later section of this paper.

*1.2. Potential Impact of Pornography Viewing at a Young Age*

The potential impacts of early exposure to pornography, whether accidental or deliberate, on sexual attitudes and morals are problematic [13]. An early first encounter with pornography can impact young people behaviourally and psychologically through experiencing hyper-sexualisation and emotional disturbance, such as reinforcing and facilitating addiction and having unrealistic sexual beliefs and attitudes [13,18–20].

Further, there is an increasing body of research that has identified a link between pornography viewing and sexual harassment or sexual violence, as well as the perpetration of sexual assault (e.g., [21–24]). This association between young people's pornography use and sexually aggressive behaviours [5,25,26] can be understood by the observation that pornography viewing may strengthen attitudes supporting sexual violence. Theoretical frameworks which can explain this association are social learning theory [22] and perceived realism [27,28]. Perceived realism understands media content, such as pornography, as real life.

Another concerning outcome of young people's pornography viewing is that simulated gender roles may be generalised to real-life sexual relationships [20,25]. Pornography presents sexual scripts that highlight aggressive relationships, objectification of women, and rigidly gender-stereotypical roles [25]. These scripts often portray women as submissive sex objects, and men as dominant and aggressive [29–31], and subsequently perpetuate gender inequality [13]. Photoshopping and video editing, combined with excessive highlighting of thin female actors and muscular and well-endowed male actors in pornography, can negatively impact young people's body image [32].

Despite these risks, pornography is often viewed by young people as a source for sexual exploration, to understand sexual identities, as a form of sexual play [33] or for sexual education [14,20]. Often young people turn to pornography as a source of education that they may not be receiving elsewhere [20]. McLelland [34] proposes that young people may be actively choosing, rather than being coerced into, engagement with online pornography to explore new ways of negotiating and interacting sexually to enhance sexual exploration and help in developing healthy sexual identities. These proposed benefits create an environment in which pornography may be useful for young people. However, this environment needs to be analysed using a critical lens that young people may not yet possess. The development of this critical lens may help to limit some of the real harms associated with early pornography exposure listed in the paragraph above.

*1.3. Parental Awareness and Responses to Pornography*

Given the documented harms and the easy and affordable access young people have to pornography and their lack of a critical lens, the roles parents and carers play in mediating the relationships between young people and pornographic consumption are important to consider. However, generational dissonance around pornography may produce closed communication about this taboo topic [35]. This can impact both young peoples' levels of openness in sexual education discussions with parents and parental underestimation of their child's pornography exposure [36].

This level of openness can be impacted by parental unfamiliarity with their children's exposure [36,37]. One study found that only 24% of parents thought their child had or was very likely to have been exposed to pornography [11]. Similarly, 80% of parents were unaware of their child-reported exposure to pornography in a sample of Taiwanese and Chinese schools [37]. Additionally, parents often overestimated the likelihood of their children confiding in them about sexual education and pornography [12].

Often parents feel these conversations should not happen with children who are too young [16,34,38]. The study by Robinson, Smith and Davies [16] found that parents believe

that primary school children are too young to engage in critical conversations regarding pornography. However, other parents argue that any sense of 'innocence' that parents are trying to prolong is misguided as sexuality is very much part of young people's development [16]. Despite these contrasting viewpoints, parents feel responsible for educating young people about online pornography; in one study, 77% of parents felt this was their full responsibility [11]. However, Pacheco and Melhuish [39] found that four out of ten parents do not know where to access resources to support these types of conversations with children. These findings suggest that this "taboo" topic might benefit from normalisation to facilitate supportive and open family communication. As many parents feel that they are responsible for educating their children about pornography, it would be useful if there were more accessible resources.

### 1.4. Parental Communication about Pornography

There are some inconsistencies in the literature regarding parental communication about pornography. Studies found some parents and carers are unlikely to raise the topic of pornography viewing due to their own embarrassment [11,12]. As a result, this could inhibit close and open communication and, in turn, impacting on the information young people are receiving about sexuality and pornography. Conversely, the same study found that 61% of parents spoke honestly with their children about the pornographic material they had seen, and 29% of parents increased monitoring and supervision of their child's online activity after pornography exposure [11]. However, mediating communication after their children's pornography exposure may be leaving it too late. These findings suggest that parents may need more support in responding to their concerns about the ideal time to commence communication.

Parental communication has been shown to have an impact on young people's pornography use. For example, one study [40] found that communication was poor between young people and parents about potentially online risky behaviours, such as sexting [41]. However, when parents consistently communicated to young people about pornography, young people's attitudes toward pornography were significantly less positive [17]. Conversely, another study suggested that when young people felt their parents to be especially controlling, they used pornography more often [42]. Another study [22] further suggests that poor emotional bonds with caregivers also predict sexually aggressive behaviour. Therefore, emphasis on effective and open communication and supportive connections between young people and their caregivers could help increase educational opportunities and limit the harmful effects of pornography use.

### 1.5. Educational Support about Pornography Usage

While parents see themselves as the primary resource for information, young people often look elsewhere for sexual education. Young people rank school second, after peers, as a preferred educator about sex and sexuality, even though many received no education about internet pornography as part of sex education [12]. Some young people also reported that school education could be negative, gendered and heterosexist [43]. Additionally, school-based sexual education has been seen as an embarrassing topic for young people, with teachers expressing reluctance to teach the topic [43] and students expressing concerns about teachers providing sexual education rather than external providers [12]. While parents and young people have expressed strong support for education about online pornography to be carried out in formal school sexual education [28,36], this education needs to be delivered in conjunction with parent education. This is captured by the whole school approach [44,45], which posits that the most effective way to educate young people is by involving all members of the school community, including educators, students, and parents.

However, education for young people about sexual health and, specifically, pornography use appears to be outdated [28]. School-based sexual education needs to be more relevant and responsive to the needs of young people, placing importance on media liter-

acy and referring to issues that may arise from pornography viewings, such as safer sex practices, body image and self-esteem, and attitudes toward the opposite sex [16,28,36,46]. Furthermore, in order to best assist parents in having these conversations with their children about the viewing of online pornography, it is critical to know which information is most relevant to parents.

### 1.6. Gaps in the Literature

The research highlights inconsistencies regarding parental awareness of and communication about young people's pornography exposure [12,14,15]. There appears to be continued parental underestimation of and misunderstanding of young people's pornography use [35,37,39]. These gaps in parental understanding of pornography use are problematic because of the unmediated ideas about, for instance, consent and sexual violence and the potentially harmful impacts of pornography viewing. Unmediated ideas are concepts that are presented to young people without having educational conversations that encourage young people to more fully understand what is being presented. As a result, the role of open and honest communication between parents or teachers and young people is critical. Further research is needed to understand how equipped parents feel in educating and communicating with young people.

The present study aims to highlight and investigate the education and support needs of parents and carers of young people. A descriptive rather than inferential approach is taken to explore parental beliefs about the content and the support that is needed in mediating the relationship between young people and pornography consumption. The study investigates parental experiences and beliefs about how young people are educated about pornography viewing in Australia. The strong social mood about issues of consent, body image, and mental health necessitates an understanding of how parents view pornography education in terms of when, where, and who should be educating young people about sexuality, whether that includes themselves, school programs, teachers, or a combination of these support systems.

## 2. Method

To investigate the identified aims, data were collected through semi-structured online interviews. The online interviews with parents of school-aged children (5–18 years old) elicited their perspectives on what supports are needed for young people in response to pornography viewing.

### 2.1. Participants

Participants were recruited via advertisements through Facebook groups and Reddit communities with targeted interests in parenting, education, or family in the Australian population. A link was provided in these advertisements, which screened for eligibility. If eligible, the participants completed an online survey and then had the option of signing up for an interview. Subsequently, a paid Facebook advertisement on the Deakin University Research page was used. To increase recruitment for the interviews, a direct link to register for the interview was also posted. Participants were eligible for recruitment if they had at least one child aged 5–18 years old, currently enrolled in school.

Nine parents agreed to be recruited for the study; however, one participant was excluded prior to participating as their child did not meet the age range. Participants (*n* = 8) included six mothers and two fathers currently residing in Australia who collectively had five daughters, thirteen sons, and zero non-binary children. The daughters were aged between 5–20 years old, and the sons were between 6–19 years old. These figures are presented in Table 1 with pseudonyms used to protect confidentiality.

**Table 1.** Demographics of Participants and Their Children.

| Participants (Pseudonyms) | Gender | Children | |
| --- | --- | --- | --- |
| | | Gender | Age |
| Elizabeth | Female | Male | 6 |
| Paul | Male | Male | 17 |
| | | Male | 19 |
| Sharron | Female | Male | 13 |
| | | Male | 15 |
| | | Female | 17 |
| Julie | Female | Female | 10 |
| | | Male | 12 |
| Naomi | Female | Female | 5 |
| | | Male | 8 |
| James | Male | Male | 15 |
| | | Male | 17 |
| | | Female | 20 |
| Nancy | Female | Male | 7 |
| | | Female | 10 |
| Jess | Female | Male | 7 |
| | | Male | 11 |
| | | Male | 14 |
| Total | F = 6 M = 2 | F = 5 M = 13 | F mean age = 12.4 M mean age = 12.4 |

F = Female, M = Male.

## 2.2. Procedures

Ethical approval was granted by Deakin University HEAG-H (Ethics approval number: HEAG-H 4_2022) and met the requirements of the Australian National Statement on Ethical Conduct in Human Research 2007 (updated 2018). Once participants were enrolled for the interview, they were invited to nominate a meeting time and provide consent for participating in the study. Interviews lasted around 20 min and were conducted online via Zoom with both the participant and interviewer in private spaces to allow for uninhibited and open communication. The interview was semi-structured, with time dedicated to building rapport between the participant and interviewer. The semi-structured nature of the interviews allowed for flexibility in the exploration of what the researcher interpreted as important to the participant [47].

The interview schedule (Table 2) is related to the support needs of young people in pornography education. It investigated general pornography viewing awareness, parent-child conversations, age of education, the context of education, and school-based sexual education through the lens of parents. Participants, in response to open-ended questions, communicated their meaningful opinions on young people's pornography support needs and educational opportunities, currently and in the future.

## 2.3. Data Analysis

Interviews were transcribed using the live transcription tool on Zoom, followed by the correction of all missed and incorrect words by the researcher. The data were edited to remove stutters (ums, ahs) and double words. Any personal identifiers were removed from completed transcripts, and pseudonyms were given to participants to protect confidentiality.

Data were interpreted in line with the ontological and epistemological foundations of a constructionist paradigm such that reality and knowledge claims are interpretations from the parental narratives [48]. Ontologically, it is accepted that participants' descriptions best reflect their interpretations of their worlds [49]. Hence, an experiential orientation was used to examine the personally meaningful ideas that reflect the subjective experience of participants' social realities [47]. The constructionist epistemology acknowledges the

researcher's active role in interpreting data to find patterns of meaningfulness emerging during the coding process [47]. Additionally, there is an interpretative process provided by the researcher involved in the thematic analysis [50], which requires the researcher to reflect on their role and impact in the thematic analysis.

**Table 2.** Interview Schedule.

| Question Number | Question | Question Probes |
| --- | --- | --- |
| Question 1 | What have you heard or read about pornography use by young people? | - Where have you heard or read that<br>- Tell me more |
| Question 2 | What conversations do you think parents should be having with their children about pornography? | - Key messages and importance<br>- Aims of key messages |
| Question 3 | What age do you think children should be educated about pornography use? | - Too young<br>- Too old<br>- Age or Criteria/Milestones |
| Question 4 | Who do you think should be educating young people about pornography use? Why? | - Parents, friends, school, online sources<br>- How responsible is each source. |
| Question 5 | Do you think education about pornography should be a part of the health education taught in schools? | - To what extent |
| Question 6 | Is there anything else you'd like to tell me about pornography use by young people? | |

Data produced via the semi-structured interviews were analysed using reflexive thematic analysis [51] to offer a rich and organic analysis of patterns across the data set to enhance the credibility of findings. First, the data were familiarised by the first author (SB), re-reading transcriptions, rewatching interviews, and note-taking on the interviewee's words. Following familiarisation, the interviews were then coded by the first author. An inductive approach was used to create codes that are reflective of the patterns of meaning in the data rather than assigning codes to pre-existing coding frames. Mostly semantic codes were used to capture the exploratory nature of the interviewee's words, with minimal latent codes used. A second coding sweep was completed by the first author, and initial themes were generated, which were then edited and refined to capture the core ideas as meaningful patterns. Themes were further refined and named following critical feedback from co-authors (MP and CS) to enhance the thoroughness and credibility of the research. Figure 1 presents a thematic map of the themes created from this study. Finally, this report was produced, and critical refinements were made following the co-authors' feedback.

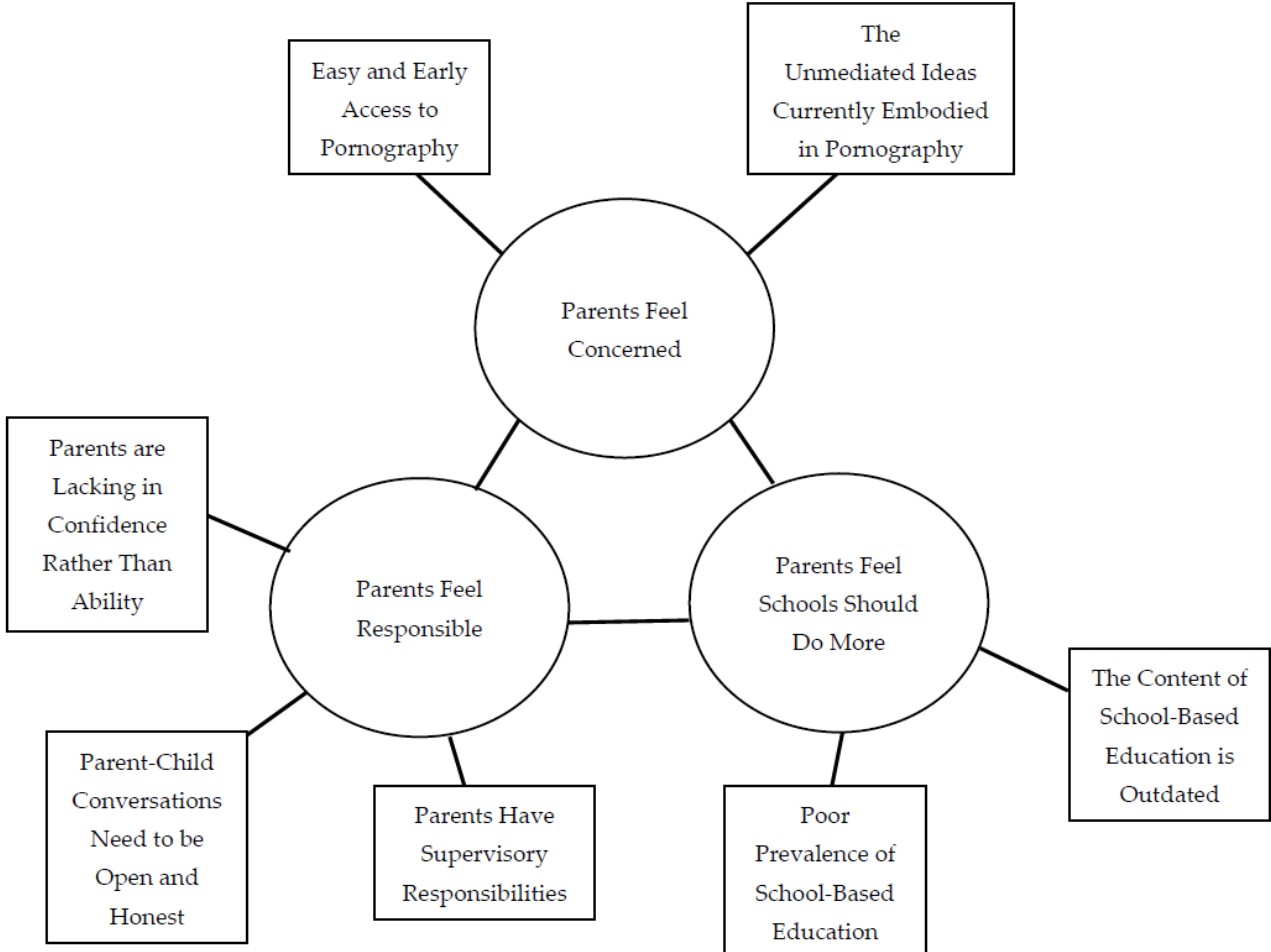

**Figure 1.** Thematic Map.

## 3. Results and Discussion

Qualitative interviews explored parental opinions and explanations of young people's pornography use and support needs. The three themes produced are presented in Figure 1, along with their subthemes. The three themes include (1) parents feeling concerned, (2) parents feel responsible, and (3) parents feel schools should do more.

### 3.1. Theme One: Parents Feel Concerned

Parents expressed general concern about their children's exposure to online pornography. In more specific terms, parents express two types of concerns which became subthemes of this general concern, (1) easy and early access that young people have to pornography and (2) the unmediated ideas that are currently being portrayed in pornographic relationships.

#### 3.1.1. Easy and Early Access to Pornography

Many parents expressed concerns that young people can access pornography easily and at a very young age. Parents suggested that this may be due to the familiarity with internet access that young people have nowadays. For example, Sharon explained:

> I know they have access, you know they're on the devices now, you know, a lot of the time. Because school is all devices as well. So, they have access, "legal access", as well as you know personal access to devices.

The prevalence of young people's connection with the internet [2] may have become even more common in the recent climate of lockdowns and online learning in Australia. In addition, Julie suggested that easy access, as well as the higher volume of pornography,

with 30 percent of internet content being pornography [52], is contributing to higher rates of pornography viewing.

> . . . through their access to the internet, I guess. So, they have more access to this pornography. And there's also just more of it out there as well. So, they're more likely to come across it.

The easy and often unmonitored access to and control over the internet that young people have correlated with the easy and hidden access to internet pornography by young people [3–7]. Access is also occurring at a young age. Most parents expressed this concern, with Julie stating, "children or kids are increasingly being exposed to pornography at a younger age, particularly through their access to the internet", and Nancy claiming, "I've heard that kids are seeing it a lot younger than what we expect. They've got access to pornography that we don't necessarily know about or understand".

These observations parallel research that found the age of first exposure can be as young as 10–11 years old [12,14]. However, parents may be more aware of the age of pornography exposure than previously thought [36,37], even if only through logical supposition. Alternatively, while agreeing that children may be seeing pornography at younger, Nancy contended that she did not comprehend pornography as a problem now:

> I guess the thing is, I don't, I don't know how much of a problem it is, maybe because my kids are really young. So I guess because I can't see the problem, I don't really know what is there to fix. So, I am like literally head in the clouds of this stuff like where I just feel like lah-lah-lah-lah-lah.

This statement highlights the discrepancies in parental responsiveness to pornography use by their children. Nancy, with children currently aged seven and ten, may not truly be aware that her children are close to the range revealed in research on the age of first exposure.

A related concern of parents is that pornographic exposure can occur involuntarily. Greater internet access [2] results in increased opportunities to be exposed to pornography by accident. Parents expressed concerns about this involuntary exposure. James explained that "you know, the online world where any porn comes at us as much as you might go after it sometimes", and Naomi stated, "it can happen without meaning to so that makes me nervous". Previous research also suggests many young people are viewing pornography accidentally [11]. Involuntary access to pornography is worrying, especially when viewing without mediation and/or education. This may result in young people associating pornographic sexual activities with real-life sexual activities. Other consequences involve the harms associated with young pornography viewing, such as sexual aggression and hypersexualisation, which raise concerns when young people view pornography involuntarily.

In response to this rapid and young access, some parents counter exposure by limiting access to the internet. This was articulated by Nancy, who said:

> . . . In my house, we have a laptop, and both me and my partner have phones, but we have no iPad and the kids don't have any of their own devices. And sometimes I think that we've been very lucky to just kind of not have to engage with that . . . I think my kids access to it is quite limited.

However, this limitation to screens in the child's household may not mean total limitation, as children can source access from other places, such as friends' houses [6].

### 3.1.2. The Unmediated Ideas Currently Embodied in Pornography

A second concern that parents expressed are the unmediated ideas that pornography presents to young people. Parents included in this study conveyed multiple concerns about the nature of pornography and how it could have damaging effects on young people. The ideas presented centre around (1) reality and fantasy, and (2) positive and negative pornography.

Reality and Fantasy

Parents expressed that it is important for young people to understand that pornography does not represent real-life relationships. Pornography depicts "fantasy" sexual relations that can be violent, abusive, and disrespectful, whereas (ideally) real-life relationships should be loving, kind, and respectful. This is highlighted by Naomi:

> So over here you've got the bubble of, you know, this is a loving, intimate relationship, you know, it's respectful, it's caring, it's nurturing, it's fun, its kind, you know, and this is real. And then you've got, over here, you've got porn, which is like not real.

This poses implications for how young people act in their own emerging sexual relationships. Parental concerns described the concerns around children replicating elements of pornographic behaviours. Jess expressed her concern about having three boys and the impact of this perceived realism of pornography: "I kind of look at them and go, that's not how I would want them to engage in sex . . . I feel like it's not mutual and it's not respectful in a lot of ways".

For young people to understand that pornography differs from real-life relationships, parent–child conversations would be useful. However, if not conversed about, parents expressed concerns about young people lacking a critical and interrogative lens backed by education about what they are consuming. Nancy explained: "If they see something related to pornography or pornography online, are they asking, 'what did that woman have to do for that' or 'what did that guy', you know, 'how much was that person being paid?'". This tension between the real and the simulated relates to previous research.

When young people try to replicate what they see in pornographic relationships [22] or perceive pornographic relationships as real [27,28], it can have harmful effects on real relationships by supporting sexual violence and aggressive behaviour [5,22,26]. The mediating role that supports systems, including parents, play to allow young people to understand the difference between simulated and real relationships and can shape how young people view and analyse pornography.

Negative and Positive Pornography

Pornography can be viewed as possibly producing a mixture of negative and positive outcomes. Parents have expressed that pornography can be used positively for sexual exploration and interest. Paul explained, "it's possible for sexuality to be portrayed in a way that's arousing and serves that purpose of being an aid to sexual expression, which is not exploitative". Further, Julie stated that pornography "can play a really important purpose. And I think it can help kids to become educated, to explore their sexuality". Previous research also described positive sides to pornography, involving sexual exploration, forming of sexual identities, sexual play or sexual education [14,20,33,34].

However, there also are concerning aspects to pornography. If young people aren't educated to critically analyse pornographic relationships, they may replicate or glamorise concerning aspects of pornography. Pornography often presents relationships with gender imbalances. Jess stated that "It's always pleasing the man, and everything's around, centred around the man in pornography", and Julie offered, "Quite often it's very like the power relationship between genders and between men and women is not necessarily healthy and exploitative as well". The consequences were explained by Sharon:

> So you don't want these 15 year old boys thinking that's how you treat women . . . It's not what you do to a 15 year old girl . . . That kind of an education, [it's] alright if you want to . . . get excited about but it's not actually how you respect women.

This gender imbalance and objectification of women has been a concern of pornography in previous research, explaining that the mediatised gender roles can be generalised to real-life relationships [20,25]. The continued perpetuation of women acting to please men

that are presented in pornography could have damaging effects on how women see themselves and how they engage in sexual relationships. Additionally, the perfectly sculptured actors that a large portion of pornography presents have been found to negatively impact young people's body images [32].

Additionally, pornography can enact violence and exploitation. Parents in this study raised concerns about the types of pornography being viewed and the impact this could have, describing these forms of pornography as "really shitty, violent, and disgusting" (Nancy), with "people being exploited, or disrespectful things being out there" (James). Paul explains the consequences: "without adult contextualisation and guidance . . . a child could have all kinds of responses that might not be great responses".

Previous research found that many young people have viewed paraphilic or similar sexual activity, such as child pornography and sexual violence in pornography [53,54]. The concerns raised by parents highlight an issue with the genres of pornography that young people are viewing and how they interpret them. Without guidance and examination, young people may interpret pornography in harmful ways. Unmediated ideas portrayed in pornography can have harmful effects, such as the potential association between young people's pornography consumption and the nature of their personal relationships [5,25,26]. While there are positive sides to pornography, adult guidance is needed to encourage healthy sexual exploration and play rather than harmful or exploitative sexual acts.

### 3.2. Theme Two: Parents Feel Responsible

This theme captures the idea that parents feel responsible for educating and conversing with young people about pornography. It encompasses three subthemes (1) parent–child conversations should be open and honest and (2) parents have supervisory responsibilities, and (3) a lack of knowledge may be better categorised as lacking confidence in pornography education. Every parent in this study highlighted their beliefs that parents have a responsibility to educate their children about pornography. Despite recognising that some parents may refuse to contemplate the possibility of their children viewing pornography [39], parents in this study attest to, at least, partial responsibility for educating their children. Elizabeth states, "It's gotta be parents, but it also, it has similar messages have to be fed in through the school as well", and James agrees, "I do think parents have a responsibility to have those conversations with their kids". When parents did not initially understand when and what to address in conversations, they wanted guidance and support in negotiating these conversations.

Additionally, the ideal age of conversational exposure was addressed, with many parents believing this occurs through maturity and curiosity rather than chronological age and that this is based on parental choice. James explains that "I don't know whether it's an age thing, like when you're seven you find out about x, when you're nine you find out about y, or whether you need to do it on a, on an individual maturity basis" and Elizabeth states:

> It's going to be a matter of looking at my son, working out where he is at, if he's starting to get interest, if he's on YouTube, if he's asking questions about porn, then that's the time that you have to follow your children's lead.

Paul adds, "it's not a one-time conversation it's the ongoing dialogue guided by the development of the developmental stage your child". This age of education should ideally line up with the age of potential exposure to pornography viewing. Inconsistencies previously detailed around the age of exposure to pornography [12–15] may be contributing to why parents avoided chronological age for initiation of conversations. This may be suggestive of a more individual evaluation about the onset of education where parents use cues to assess where the child is in terms of maturity when initiating conversations about pornography.

However, different parental opinions were about how, when or what to provide education on; parents agreed that they have a responsibility to educate young people about pornography. This research replicates existing findings about parental communications

that, whilst some parents are unwilling to raise the topic [36], most are thinking about how to communicate about pornography [11].

### 3.2.1. Parent–Child Conversations Need to Be Open and Honest

When investigating which conversations parents thought that they should be having with their children around pornography, the important themes were openness and honesty and not pretending that pornography doesn't exist. Julie stated that conversations must recognise that "pornography is there, it's available and not kind of sweeping it under the carpet. Additionally, they do not make it something that they are forbidden to do or access. So just having a really open conversation around pornography". James expanded with, "maybe it could do with being demystified and have some of the secrecy about it brushed aside, and have people speak honestly and openly about how it works, how would it be addictive and how it can be beneficial". Further, parents suggested that conversations need to become normalised. These conversations should occur without fear or discomfort.

> I think the pornography part is still the taboo part ... we've come a long way in being more open about talking about sex ... But I think that ... pornography in itself is probably something that's still not talked about, people don't admit to watching it.
>
> (Jess)

Two of the parents were very suggestive toward the idea of having more conversations about the dominant themes of pornography whilst not talking specifically about pornography.

> Every single moment is a teachable moment. You know, so like if a song comes on the radio and it's talking about slapping my bitch whatever, I don't know, maybe that's a teachable moment. Like what is this person trying to say...everything is a teachable moment if you just put it in the right context.
>
> (Nancy)

Sharron observed that education about these themes "takes a lot of people ... , a lot of different inputs ... to mould the kids so any ... conversations about it are positive conversations aren't they?"

Overall, parents all made very rounded and positive suggestions regarding conversations with young people about pornography. Previously it has been found that there is generational dissonance around pornography that can impact young people's willingness to be open in sexual education discussions [35,36]. However, parents in this study support the idea that consistent, regular and open parent–child two-way communication about pornography can lead to more critical analyses by young people of pornography. This indicates that parents may have the tools and ideas to effectively communicate about pornography, and barriers may be related to parents lacking confidence instead. Open, honest, and age-appropriate communication from parents can create positive environments for young people to explore their sexuality whilst limiting the harms of pornography.

### 3.2.2. Parents Have Supervisory Responsibilities

Most parents agree that pretending pornography doesn't exist is counterproductive, as exemplified in the following statements:

> So to ignore that or pretend it doesn't happen seems very blinkered ... It would be like not talking about drinking in health education, everybody drinks, but oh gosh we can't talk about drinking, its dumb. If everyone is having sex and everyone's watching porn then of course we would have to talk about it.
>
> (James)

> You can't pretend it doesn't exist. It's there, it's here. It might not be everyone's ideal, but, children, if they don't hear it from you, they go find it out for themselves.

(Elizabeth)

These comments are reflective of crafting an environment where communication about pornography is open, educative, and responsive to the needs of young people.

The parenting role spreads further than just communicating with young people about what they are being exposed to. This role also involves supervision of access to pornographic content. Several parents believe that controls or limited access to screens are necessary so young people can't access the harmful sides of the internet. Sharron explained, "I kind of am a big believer in like the home is where you have to set the boundaries for ... the amount of time kids spend online and ... what sites they are allowed to look at". Controls and limited screen access are just two ways these parents suggest controlling the harmful side effects of pornography.

However, some parents expressed that they are not fully aware of what their children are seeing and how they are seeing it, suggesting that such controls on screen time may be largely ineffectual. Nancy expressed: "I just don't know what they're seeing. I don't know how they're accessing it ... I am maybe a bit of a luddite with this stuff ... I really have no idea how people access this stuff", and James stated, "It just wasn't something on my radar ... it's not until you hear your kids say, I saw something that you say oh yeah, they do, they see that stuff". Previous research has also found that parents are often unaware of their child's exposure to pornography [36,37].

The supervisory responsibilities presented by parents have been replicated in previous research showing that parents use restrictive mediation for young people [6] or may increase mediation in response to known pornographic viewing by young people [11]. While controls and limited screen time are one type of restrictive mediation to limit young people's access to the internet and pornography, creating an open and honest environment of responsive communications, as described in the last section of this paper, may be more beneficial in the longer term to give young people the education they need to explore their sexuality safely.

### 3.2.3. Parents Are Lacking in Confidence Rather Than an Ability

Parents feel responsible for educating young people about pornography in this study and in previous research [11,16]. Currently, many parents expressed that they do not feel equipped to do so. Concerns were raised about parents lacking the language and resources around having conversations and providing empowering educational opportunities for young people. Nancy explained the dilemma with, "I just sometimes feel like I sometimes don't have the answers and I go on these absolute rant monologues that she's just like oh my god so boring", and Naomi said, "I wouldn't be surprised if other people in my generation feel like we lack information and language". Additionally, Nancy suggests concerns about overstepping in these conversations: "I feel like if I ask too many leading questions ... I'm looking for the answers". Previous research also found that parents lack access to resources that provide guidance on how to educate young people about pornography and online safety [6,39].

However, parents in this study presented sound ideas around communication styles and supervisory responsibilities for young people, which suggests they are not ill-equipped to educate young people about pornography. Both Naomi and Nancy were able to identify that the age of first pornography exposure is younger than generally expected and that education and open conversations with young people about pornography is needed. In particular, Nancy states that she wants "all kids to know that that's [pornography] not reality", suggesting that she has a sound idea of what young people need to understand in relation to pornography. The strong foundations participants presented about ways of engaging in pornography education, communication, and nuanced demarcation of the harms associated with pornography indicate that parents potentially are lacking in confidence rather than lacking in ability or understanding. For example, Paul explained that the contrived nature of pornography produces a "colonisation of what in my mind would ideally be a process of personal exploration and growth that's really influenced by

an industry". This indicates that Paul has a critical understanding of the economic goals of pornography producers that allows for informed communications regarding the perceived realism of pornography by young people. Potentially, parents may not need more tools and education to better communicate about online safety and pornography concerns but may need better recognition that they have ideas conducive to having confidence in their ability to educate and communicate.

### 3.3. Theme Three: Parents Feel Schools Should Do More

This theme captures how parents feel about the current education or lack thereof, that schools are providing to young people in terms of pornography. It encompasses two ideas; that there is a low prevalence of school-based education and that such education is often outdated for young people.

#### 3.3.1. Poor Prevalence of School-Based Education

When turning to discuss the roles that schools play in pornography education, many parents believed that sexuality education and pornography education are either absent and/or deficient. Sharron expands on this by saying that teachers "don't really delve into the whole pornography thing at school and my kids are in the Catholic school system, so they don't really, they don't touch on pornography at all . . . so there is no educational there is no discussion, unless the discussion is at home".

A majority of the parents agreed that the consumption of pornography by young people could not replace a comprehensive sex education program. Jess explained: "I don't want them getting their education about sex through pornography. They need to get it through school, they need to get it through their parents, they need to talk about it, they can talk to their friends". Further, the easy access to pornography led Paul to make the point that for some young people pornography "actually takes the place of sexuality education in so many cases". These narratives suggest strong parental support for the use of school time to include a sex education program as an important part of the school curriculum in order to counterbalance the messages that are being consumed by children through pornography.

Additionally, despite these concerns about deficiencies in the current school-based sexuality education, most parents believe pornography education should appear in existing health education taught in schools. Sharron exemplified this general feeling:

> I think that . . . teenagers don't like what their parents have to tell them... So even if it misses out a bit at home at least there is some capture at school . . . I think it should be part of the sex education process. Like I think that's a good home for it to kind of sit in between the sex education and the cyber and the internet and the internet education.

Moreover, parents believe pornography education is as important as other practices currently taught in health education. Elizabeth suggests, "it's no different to talking to . . . kids about drink driving and drug use", and James says, "it would be remiss of us not to include that in our conversations around sex and sexuality and responsibility and care for each other".

It has been found previously that there is general support for pornography education to be taught in schools [28,36], but this support can be dependent on location and whoever holds political power. Additionally, it has been shown that some young people are not receiving any education on internet pornography as part of their sex education [12]. Parents in this study suggest that pornography education is absent in their children's schools and that pornography is taking the place of sex education for children. For parents who choose not to educate, young people may have no critical understanding of the themes and images presented in pornography, potentially producing replication of sexually aggressive behaviour. The implication is that responsibility for pornography education must currently lie with parents.

### 3.3.2. The Content of School-Based Education Is Insufficient and Outdated

When specifically focusing on the content of school-based education, according to parents, it is perceived as being outdated, traditional and uninteresting by young people. Julie explains one of the key organisational problems eloquently:

> I asked a question whether the boys and girls were separated. And I think they might have been at some point, which, again, is something . . . that's just crazy that just speaks to the very old understandings of gender and sexuality and all that sort of stuff. [I am also] not sure about the information that they're getting, and the kids are not engaged . . . It's just something that they have to sit through. So, in that sense, they're not necessarily successful, I think.

The idea of disengaging content is also taken up by Paul:

> I feel like we do quite good reproduction education and even not bad education about sexually transmitted infections, for example . . . But I think there's just an enormous gaping chasm in spaces for young people to talk about sexuality as opposed to sort of reproduction or safety.

More specifically, Paul analysed a dichotomous split between sex education and pornography. He stated:

> I feel like there's sort of nothing in between, there's either that, which is engaging, but for terrible reasons and I think ways that can be quite harmful, and then what we do around sexuality is so dry, so oriented towards this idea of sex equals danger as opposed to sex equals pleasure. That's almost totally disconnected from the experience of young people exploring their sexuality.

This indicates a school-based preference for an education that is safe, not engaging, dry, and uninteresting. Previous research also shows that sex education in schools is lacking, negative, gendered and heterosexist [43], despite being highly ranked as a primary educational source for information about sex and sexuality by young people [12], and a highly ranked educational responsibility by parents [11]. The current school-based sex education presents ideas that will not provide young people with the ability to critically analyse what they are seeing in pornography and how pornographic relationships can be unhealthy and damaging. Additionally, what is taught in relation to pornography differs so greatly across schools. Parents believe that young people need stronger educational backing to understand pornography better, which may, in turn, lead them to turn away from pornography for sexual education.

Most parents were supportive of schools educating young people about pornography in some way. Specifically, Jess discussed how the school created a guide for her to follow in the conversations she had with her children about sex and sexuality.

> I didn't know I was kind of not sure how much to give them how much to hold back and so I think being guided by the different sessions that the school would hold, and you being a part of that just helped, and you were kind of like okay, that's enough for now because you kind of went yeah that's probably all they need to know at this age.

Additionally, Jess suggested that having the school employ external sources to teach sex education was very helpful to "open the conversation by having someone who's actually equipped to, to start that conversation was a better option for us". Previous research also suggested young people preferred external sources over teachers providing sexual education because of reasons of embarrassment [12]. However, external providers also teach pornography education with different program orientations and at different levels. Jess' understanding also provides support for external providers but on the basis of expertise.

Most parents are in support of pornography education and updating the sexuality education currently being received by young people, supporting the philosophy of making

school-based sexual education more relevant and responsive to the needs of young people [16,28,36,46]. This update could provide an additional support system, on top of parents, that gives young people additional skills to assess the content they see online, which could limit the harmful impacts and help promote the beneficial aspects of pornography.

### 3.4. Limitations

Limitations of this study were that there were a small number of participants and that the study was completed with a largely female sample. Previous findings have indicated that young people are more likely to discuss sex and sexuality with their mothers [12,55]. Future research should examine parental confidence, ability, and communication style across a larger and more gender-balanced sample via both quantitative and qualitative methods to understand if parents differ according to gender in response to their children's pornography viewing. The tension between how parents feel about their inability and their actual ability may be due to personal histories with limited pornographic education or to the lack of communication provided by schools to parents about what they are doing with school children in their existing school-based programs. Further investigation is needed to unpack all aspects of parental lack of confidence and deal with this lack of confidence with a whole-school approach to pornography education.

## 4. Conclusions

This is the first qualitative Australian study to investigate parental beliefs about the education and support needs of young people in relation to pornography. As such, the current study adds to knowledge by presenting a qualitative account of parental understanding of pornography use by young people. Specifically, this study found that whilst parents feel educational conversations are important and they are willing to engage with young people about pornography, they lack the confidence to execute these conversations. This suggests that it could be useful to provide resources and support for parents to boost their confidence and encourage their communication with young people about pornography. This study also supports previous findings regarding concerns about pornography, parental responsibility, and a deficient educational system for young people around pornography. Additionally, this research highlights the need for stronger communication and agreement between schools and parents about pornography and sex education, its delivery, and how parents and schools can both support young people, as advocated by a whole-school approach [44,45], like the proposed practice framework demonstrated in the article by Crabbe and Flood [56].

**Author Contributions:** Conceptualization, S.B., M.P. and C.S.; Data curation, M.P. and C.S.; Formal analysis, S.B.; Investigation, S.B.; Methodology, S.B.; Project administration, M.P. and C.S.; Resources, S.B., M.P. and C.S.; Supervision, M.P. and C.S.; Validation, S.B., M.P., C.S. and B.K.; Visualization, S.B.; Writing—original draft, S.B.; Writing—review & editing, S.B., M.P., C.S. and B.K. All authors have read and agreed to the published version of the manuscript.

**Funding:** This research received no external funding.

**Institutional Review Board Statement:** Ethical approval was granted by Deakin University HEAG-H (Ethics approval number: HEAG-H 4_2022) and met the requirements of the National Statement on Ethical Conduct in Human Research 2007 (updated 2018).

**Informed Consent Statement:** Informed consent was obtained from all participants included in the study.

**Data Availability Statement:** Data sharing is not applicable in this article.

**Conflicts of Interest:** The authors declare no conflict of interest.

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
