# Peer review of "“It’s Not a One-Time Conversation”: Australian Parental Views on Supporting Young People in Relation to Pornography Exposure"

_psych, doi:10.3390/psych5020034_

Round 1

Reviewer 1 Report

This is an interesting paper about an important topic. It is not overly clear to me how this paper links with the scope of the journal so I would suggest some clarification is needed in that respect. Some minor points for consideration below:

1.     The title should note that the study is with Australian parents e.g. Australian parental views…

2.     Abstract:

-       I would suggest replacing ‘poses a threat’ with ‘has possible negative implications’

-       Authors describe a ‘qualitative group study’ but in fact the methodology reported later was individual interviews. This should be corrected.

-       The study is described as a ‘pilot study’ in the abstract and elsewhere. It is not however clear what it is a ‘pilot’ for. My understanding is that this is a qualitative study and not a pilot. If this is the case, it should be named as such and all references to pilot study removed.

3.     Introduction:

-       Lines 46-49 make some rather bold statements about the impact of pornography, however, the statements are only supported by a narrative review. I would expect to see primary research supporting such statements, otherwise it can mislead.

-       Authors refer to a ‘growing body of research’ linking pornography and sexual violence, however only one study is cited. More citations needed at this point or rewording of sentence.

-       Typo at line 160

4.     Method

-       The authors refer to ‘cross-sectional semi-structured interviews’. There is no such methodology. Remove the cross-sectional part.

-       Lack of clarity here regarding what the data collection involved.

5.     Results/Discussion

-       This is quite a long section and the integration of other research tends to get lost on the data. If the authors wish to retain this amalgamation of results and discussion I would suggest adding an implications section (and drawing out the parts of this already reported) – what are the implications of these findings for further research and considerations for practice.

Reviewer 2 Report

Thank you for asking me to review this interesting paper. Overall, I think that the study has potential to make a contribution to this literature. While I am primarily a quantitative researcher, from what I can tell the study was carried out competently. Given that pornography is such a politically charged topic it is impossible to talk about the potential effects of pornography in a way that would please all readers. I do think this article has a tendency towards privileging the negative effects paradigm somewhat uncritically (e.g., the first paragraph of section 1.2 could be described as alarmist. Why is having permissive sexual attitudes a negative thing?), but some discussion of potential positive effects are given, which is good to see. Another example is in section 1.4. The line “However, when parents consistently communicated to young people about pornography, young people’s attitudes about pornography were significantly less positive” seems to have an implicit assumption that the correct way to think of pornography is as being bad and that making children think less positively of pornography should be our goal. I think a more neutral position is that we want to help children who consume pornography to be critical consumers who think about the ways in which pornography does or does not reflect real world relationships.

The term “unmediated ideas” is used throughout. I think the authors might mean something like “unrealistic messages”, but it was never quite clear to me. The authors may consider rephrasing (or at the very least defining this term) for clarity. “Mediation” is another term that is used in a bit of an ambiguous way (and “mediation” has a particular statistical meaning which may further confuse the issue), e.g., “parents use restrictive mediation for young people” (line 520) is unclear to me. Again the authors might need to rephrase or consider defining this term.

Some other observations are outlined below. All of these are relatively minor.

The discussion of the first theme seems to repeat research outlined in the introduction (e.g., statistics around age at first exposure). I would consider cutting this back to make the piece more concise.

Line 38. Is the suggestion that parents are reading this literature and it is confusing them? This seems unlikely.

Line 56 – perceived realism is not really a theoretical framework (at least not in the same way that social learning theory is). You may need to give more context for this for the reader, as just the phrase “perceived realism” would probably not mean much to those not familiar with the pornography effects literature (and perceived realism is referred to a few more times later in the paper).

Lines starting 155 and 157 repeat info.

Line 196 – You may want to clarify that this is the Australian national statement

Line 297 – The claim that this is likely “avoidance” seems somewhat judgmental of Nancy. I don’t think this was your intention, but you might consider rephrasing this paragraph.

Line 389. The reference to “underage women and girls” seems a little out of leftfield. Was this a concern being articulated by participants? If so, the authors may want to contextualise this a little more.

Line 481 – There is a statement that conversations lead to “less positive perceptions”.  As above, I think that phrasing is a bit loaded. Also it seemed to me your participants were talking more about reducing the negative effects or porn, not the degree to which porn is perceived as positive or negative per se.  

Section 3.2.3. The “ability but lack of confidence” aspect didn’t came across to me in the illustrative quotes chosen. For example, Nancy’s quote relates to lacking information/language to discuss this issue, which seems to be about a lack of ability. The authors may want to consider whether they need different quotes/explanatory statements to better illustrate this theme.

Line 565 – The quote from Paul is interesting, but it seemed out of place in that paragraph in that it doesn’t really connect to the notion of schools’ role in educating students about pornography. The authors may want to consider moving this quote or elaborating on how the quote connects to the topic covered in that paragraph.

Line 589 – This might be too definitive a claim as this really depends on country/education system. For example, Maree Crabbe has an article in American Journal of Sexuality Education around a sex ed program for porn literacy in Australia.

Throughout the manuscript the study is referred to as a “pilot study”, but it is never really made clear what is being piloted.

A few typos here and there but the writing is generally clear. 
